# Identification of Powdery Mildew Resistance-Related Genes in Butternut Squash (*Cucurbita moschata*)

**DOI:** 10.3390/ijms252010896

**Published:** 2024-10-10

**Authors:** Yiqian Fu, Yanping Hu, Jingjing Yang, Daolong Liao, Pangyuan Liu, Changlong Wen, Tianhai Yun

**Affiliations:** 1Beijing Vegetable Research Center (BVRC), Beijing Academy of Agriculture and Forestry Science, Beijing 100097, China; fuyiqian@nercv.org (Y.F.);; 2National Engineering Research Center for Vegetables (NERCV), State Key Laboratory of Vegetable Biobreeding, Beijing Academy of Agriculture and Forestry Sciences, Beijing 100097, China; 3Vegetable Research Institute, Hainan Academy of Agricultural Sciences, Key Laboratory of Vegetable Biology of Hainan Province, Hainan Vegetable Breeding Engineering Technology Research Center, Haikou 571100, China; ziy2013@163.com (Y.H.);

**Keywords:** powdery mildew, pumpkin, DEGs, *CmMLO*, transcriptome, S gene

## Abstract

Powdery mildew infection is a significant challenge in butternut squash (*Cucurbita moschata*) production during winter in Hainan, China. The tropical climate of Hainan promotes powdery mildew infection, resulting in substantial yield losses. By utilizing transcriptome and genome sequencing data, SNPs and potential genes associated with powdery mildew resistance in butternut squash were identified. The analysis of differentially expressed genes (DEGs) following powdery mildew infection revealed several genes involved in resistance, with particular focus on a resistance (R) gene cluster that may be linked to the observed resistance. Two *MLO* genes in clade V from *Cucurbita moschata* may not be directly associated with resistance in the two genotypes studied. These findings are expected to contribute to the development of genetic tools for improving powdery mildew resistance in Cucurbita crops, thereby reducing yield losses and enhancing the sustainability of butternut squash production in Hainan and other regions.

## 1. Introduction

Butternut squash is a genotype of *Cucurbita moschata*, known for its robust branching habit and hammer-shaped fruits with a sweet, nutty flavor, making them suitable for roasting and steaming. The mini-type butternut squash has gained increasing popularity in the Chinese market because of its convenience, portability, and extended shelf life. Hainan Province, located in the southernmost part of China and surrounded by the South China Sea, has a tropical climate and is the main vegetable production area, including butternut squash, for the mainland Chinese market, particularly during winter. Although Hainan experiences milder winters than other regions of China, the combination of relatively low temperatures, high humidity, and limited sunlight during this period creates favorable conditions for the outbreak of powdery mildew, which significantly affects butternut squash production. Incomplete statistics suggest that powdery mildew can lead to yield reductions of 30–70% in Hainan, causing economic losses amounting to hundreds of millions of yuan annually (Yun, personal communication). Therefore, understanding the genetics of powdery mildew resistance and developing resistant cultivars may be the most effective strategy to alleviate this issue in Hainan.

The primary powdery mildew species infecting *Cucurbita* species is *Podosphaera xanthii*. As a biotrophic pathogen, powdery mildew fungi rely on living cells and tissues of the host plant for nutrition. After invading the epidermal cells of the host, it forms haustoria, which allows it to absorb nutrients from the host plant cells [1]. Previous studies have identified multiple powdery mildew resistance genes, among which the R gene-mediated resistance response triggers localized programmed cell death in the host plant, limiting the spread of powdery mildew as a biotrophic pathogen, thereby achieving immune effects [2]. The *Mla* locus, conferring resistance to powdery mildew in barley, is believed to be tandemly arranged with three typical R genes that have an NBS-LRR structure [3,4,5]. The *SGT1* gene is an essential component mediating R gene-mediated plant disease resistance signaling pathways [6]. Therefore, mutations or silencing of the *SGT1* gene can lead to the loss of *R* gene-mediated disease resistance, resulting in susceptibility. Its overexpression can enhance HR response and improve powdery mildew resistance [7]. The *RPW8* gene found in *Arabidopsis* may enhance salicylic acid-dependent basal defense against powdery mildew by inducing programmed cell death [8]. Among all homologous chromosomes in wheat, as many as 100 QTLs related to powdery mildew resistance have been detected [9,10]. Because of the incomplete expression of some major R genes, the combined participation of minor resistance genes, and other multigenes, as well as the influence of the environment, powdery mildew disease resistance ultimately manifests as a quantitative trait [11].

In addition, previous studies on powdery mildew resistance have identified a susceptibility gene that participates in plant immune responses [12]. One of the most well-known is the *MLO* gene (Mildew resistance locus o) [13]. The loss-of-function mutant (*mlo*) forms localized cell wall deposits, preventing the powdery mildew spores from penetrating the host’s epidermal cells to form haustoria and hyphae, thereby blocking further infection by powdery mildew fungi [14,15]. In barley, *mlo*-based powdery mildew resistance mutants have been successfully used commercially on a large scale, with over half of the spring barley varieties in Europe containing the *mlo* gene, exhibiting complete resistance to powdery mildew, and this resistance has remained durable over the decades [16]. As more genomes were sequenced, it became clear that *MLO* genes are part of a large gene family, all encoding seven transmembrane proteins, and have diversified into several clades with different functions in plants [17,18]. The *mlo* genes involved in powdery mildew resistance in monocots like barley and wheat are in clade IV, while in dicots, they belong to clade V [17]. The *MLO* gene was also found to play an important role in response to PM in cucumbers and other cucurbit crops. In cucumber, a *MLO* (*Csa5G623470*) in clade V, linked to the locus *pm5.1*, has been identified as the causal gene for durable powdery mildew resistance [19]. The same loss-of-function mutation *MLO* genes (*Csa5G623470*, named *CsaMLO8*) was proven to cause hypocotyl resistance to PM [20]. The other two *MLO* genes in clade V were also cloned, and overexpression of all three *CsaMLO* genes in the tomato *mlo* mutant restored susceptibility to PM to a different extent [21].

Several genetic studies have identified loci associated with resistance to powdery mildew in *Cucurbita* [22,23,24,25,26,27]. Among these, the *Pm-0* locus, which was considered introgressed from a wild *Cucurbita* species, provides resistance to powdery mildew in commercially cultivated *Cucurbita* varieties [23,24]. This locus contains several candidate genes that might together contribute to resistance [23]. Another major dominant locus *CpPM10.1* in zucchini contains an *Arabidopsis* RPW8 domain after fine mapping [27]. The *MLO* genes in clade V of *Cucurbita* species were identified and shown to exhibit differential expression between resistant and susceptible plants [28,29]. In this research, we aim to identify the possible powdery mildew resistance-related genes from transcriptome data after powdery mildew infection and to analyze the overall diversity and SNPs of the two genotypes of squash using genome sequencing data. Reliable SNPs identified within and between the two butternut squash genotypes can be utilized for genetic mapping. Analyzing differentially expressed genes after powdery mildew inoculation in the two squash genotypes will enhance our understanding of the genes involved in powdery mildew resistance and provide genetic tools for improving resistance in *Cucurbita*.

## 2. Results

### 2.1. Evaluation of Powdery Mildew Resistance of Cucurbita moschata

The genotype ‘YD26’ exhibited significantly lower incidence and severity of powdery mildew infection compared to ‘SF02’ (Figure 1A). The leaves of YD26 showed minimal symptoms, with only sporadic occurrence of white powdery spots observed on leaves and stems. In contrast, SF02 plants displayed widespread infection, with prominent white powdery lesions covering substantial portions of leaf surfaces

Throughout the experiment, YD26 plants maintained vigorous growth with minimal impact from powdery mildew. Conversely, SF02 plants exhibited stunted growth, leaf chlorosis, and reduced vigor because of severe powdery mildew infestation. Statistical analysis of disease scoring between YD26 and SF02 on day 10 (240 hpi) confirmed the significant differences in powdery mildew resistance (Figure 1B). The average infection score for 14 YD26 plants was 1.86, while for 16 SF02 plants, it was 3.75. A significant difference in the disease index was observed between YD26 and SF02, with YD26 exhibiting notably higher resistance, as confirmed by a *t*-test analysis (*p* = 0.00077).

### 2.2. Variations in the Whole Genome and Transcriptome Sequencing

SNPs and InDels variations derived from whole genome and transcriptome sequencing data are provided in Appendix A. Comparative analysis of SNP mutations between YD26 and SF02 revealed distinct genomic profiles, as illustrated in Appendix A. YD26 displayed 94,910 heterozygous SNP sites alongside 481,678 homozygous SNP sites, resulting in a heterozygosity ratio of 16.46%. In contrast, SF02 had a higher heterozygosity ratio of 33.23%, with 187,969 heterozygous SNP sites and 377,686 homozygous SNP sites. YD26 showed a total of 576,588 SNPs, including 372,776 transitions and 192,879 transversions, while SF02 had 565,655 SNPs, comprising 379,341 transitions and 197,247 transversions. Both genotypes displayed similar trends in SNP mutation types, with T:A > C:G and C:G > T:A mutations being the most common, while C:G > G:C mutations were least frequent.

A comparative analysis of SNP variation across genome-wide and coding sequence (CDS) regions between YD26 and SF02 revealed both similarities and notable differences (see Appendix A). YD26 showed slightly fewer SNPs in intergenic regions (138,506) compared with SF02 (153,324) and in upstream gene regions within 5 kilobases (124,114) compared with SF02 (124,314). Conversely, YD26 had a slightly higher count of SNPs in downstream gene regions within 5 kilobases (92,413) compared with SF02 (90,169). Additionally, YD26 exhibited a higher count of synonymous coding mutations (28,843) compared with SF02 (28,228). In addition to the overall differences between the two genotypes, we conducted the following detailed analysis of genetic variations within specific potential candidate genes that might contribute to the observed differences.

### 2.3. Differentially Expressed Disease Genes during Powdery Mildew Resistance

Analysis of the transcriptome sequencing data revealed distinct gene expression patterns in both YD26 and SF02 genotypes before inoculation with powdery mildew (0 h post-inoculation, 0 hpi). A total of 716 genes showed significant differential expression, with adjusted *p*-values below 0.05 and an absolute log2FC > 0.5, as detailed in Appendix A. Of these 716 differentially expressed genes (DEGs), 344 were upregulated and 372 were downregulated, as illustrated in Figure 2A. Under non-inoculated conditions, several genes in YD26 exhibited upregulated expression compared with SF02, suggesting their potential roles in underlying molecular pathways. Notably, CmoCh18G013170, annotated as a G-type lectin S-receptor-like serine/threonine-protein kinase and homologous to *Arabidopsis* At5g24080, emerged as the most prominently upregulated gene. Additionally, within the top ten DEGs, CmoCh06G007400, identified as a putative disease resistance protein RGA4, and CmoCh16G004710, annotated as a pathogen-related protein-like, indicated potential involvement in disease resistance mechanisms.

Moreover, the gene CmoCh02G016340, annotated as a two-component response regulator-like *APRR2* and associated with fruit color regulation in several gourd species, also exhibited differential expression. YD26 showed downregulation of this gene compared with SF02. In contrast, SF02 had higher expression of top-ranking DEGs such as CmoCh18G008700 (F-box protein, *SKIP5*) and a probable leucine-rich repeat receptor-like protein kinase, homologous to *Arabidopsis* At1g35710. Gene Ontology (GO) annotation of all DEGs revealed the top 10 categories in Biological Process (BP), Cellular Component (CC), and Molecular Function (MF), as shown in Figure 2B. Top annotations include processes such as oxidation–reduction, transmembrane transport, and secondary metabolite biosynthesis. These findings show genotype-specific differences with certain genes expressed exclusively in either YD26 or SF02, indicating potential genetic variations related to the response to powdery mildew.

Thousand more DEGs compared with 716 DEGs at non-infection (00 hpi) were induced in response to powdery mildew infection in both YD26 and SF02 genotypes at 6 and 24 h post-inoculation (hpi), as shown in Figure 3A,B. At 6 hpi, YD26 exhibited a unique gene expression profile with 56.2% of its DEGs differing from the non-infection baseline (00 hpi), which is notably higher than the 30.4% observed in SF02. The total number of DEGs, both shared between the two genotypes and specifically expressed in each, increased at 24 hpi compared with 6 hpi, reflecting the greater number of genes induced as the infection progresses.

The gene expression fold change profiles are visualized in the scatter plots (Figure 3C,D). In YD26, approximately 2967 differentially expressed genes (DEGs) were identified, with around 1585 DEGs upregulated and 1372 DEGs downregulated at 6 hpi. In contrast, SF02 had 566 upregulated and 448 downregulated specific DEGs, totaling 1014. Among the 2311 shared DEGs, most exhibited similar up- or downregulation patterns between the two genotypes. However, 14 genes (highlighted in purple) displayed opposite expression patterns, including genes such as peroxidase, proteinase inhibitor, phytohormone-binding protein, ABC transporter B family member, and two-component response regulators. Additionally, 65 genes exhibited expression differences of at least twofold, as indicated by their position outside the dotted lines in the plots. Notable among these are leucine-rich repeat receptor-like protein kinase, cytochrome P450, and pathogenesis-related proteins, which are strongly associated with disease resistance.

At 24 h post-inoculation (24 hpi), both YD26 and SF02 exhibited an increased number of genes involved in defense against powdery mildew. The number of shared DEGs with opposite expression patterns rose to 45. These include genes such as MYB6-like transcription factor (CmoCh20G009160), phytohormone-binding protein-like (CmoCh16G008570), JUNGBRUNNEN 1 transcription factor (CmoCh12G002210), ethylene-responsive transcription factor ABR1-like (CmoCh04G025710), and a probable calcium-binding protein (CmoCh09G02150). Additionally, the number of genes showing at least a twofold expression difference increased to 307. These include receptor-like protein kinase FERONIA (Cmo14Ch011870), WRKY transcription factor 22-like (CmoCh16G006300), MLO-like protein 2 (Cmo18Ch008900), probable calcium-binding protein CML10 (CmoCh12G004850), ethylene-responsive transcription factor ERF011-like (CmoCh01G018820), and several leucine-rich repeat receptor-like protein kinases (CmoCh16G001430, CmoCh02G017590, CmoCh15G014460), as well as cytochrome P450s (CmoCh10G000910, CmoCh18G006850). These genes are strongly implicated in the defense response to powdery mildew.

DEGs that exhibit significant differences starting at 00 hpi and maintain either upregulation or downregulation across subsequent time points may be correlated with powdery mildew resistance (Figure 3E,F). Notably, nine specific genes—CmoCh06G007420, CmoCh06G007410, CmoCh06G007400, CmoCh06G007390, CmoCh06G007380, CmoCh15G012680, CmoCh18G001720, CmoCh18G001710, and CmoCh19G000350—demonstrated consistent expression patterns, as shown in Appendix A. Among these, a cluster of five adjacent genes on chromosome 6, spanning CmoCh06: 3,817,738 to 3,836,887 (Figure 4A), includes one cytochrome P450 gene and four disease resistance gene analogs (RGAs). These genes consistently exhibit expression in YD26. In contrast, SF02 shows no detectable expression for four out of these five genes. To further validate these findings, RT-qPCR was performed to assess the expression of these genes. Data analysis, normalized using the 2^ΔΔCT^ method, revealed significant differences in expression levels. Specifically, the four adjacent genes showed notable expression in YD26, whereas no expression data were available for SF02 (Figure 4B).

### 2.4. MLOs Family Clade V Genes in Butternut Squash

A total of 20 MLO family gene sequences were identified from *C. moschata* resequencing and transcriptome data. Based on the distribution of MLO genes in *C. moschata* chromosomes, these 20 MLO family genes were named CmoMLO1 to CmoMLO20. Among the 20 *CmoMLO* genes, they are distributed on 15 out of the 20 chromosomes of *C. moschata*. Chromosomes 2, 3, 4, 13, and 20 each contain two *Mlo* genes. The sizes of the *CmoMLO* genes in squash vary, with the number of exons ranging from 13 to 15. The CDS of these genes, which encode proteins of approximately 500 amino acids in length, are detailed in Appendix A.

Phylogenetic analysis was performed on MLO protein sequences from butternut squash, along with sequences from *Arabidopsis thaliana*, cucumber, and functional MLO genes from the following four monocotyledonous crops: barley (*HvMLO*), wheat (*TaMLO*), rice (*OsMLO*), and maize (*ZmMLO*). The protein sequences used for this analysis are listed in Appendix A. The phylogenetic tree was constructed using the maximum likelihood method following multiple sequence alignment with MEGA7.0. The MLO gene sequences were classified into seven clades and labeled I to VII. MLO genes from *Arabidopsis*, cucumber, and butternut squash were grouped into all six branches except clade IV. All functional MLO proteins associated with resistance to powdery mildew in barley, wheat, rice, and maize were grouped into clade IV (Appendix A).

In the transcriptome data of two different squash materials with different resistance levels sampled at different time points after powdery mildew inoculation, the expression levels of 20 *MLO* family genes were analyzed based on the FPKM values (Figure 5A). Among these genes, *CmoMLO7* consistently exhibited significantly higher expression compared with the other *MLO* genes in both resistant and susceptible materials. *CmoMLO7* showed a notable increase in expression at 6 h (h) and 24 h (h) post-inoculation, followed by a gradual decline after 48 h (h). In contrast, the expression levels of other *MLO* genes remained relatively low and showed no significant changes before or after inoculation. Since *CmoMLO7* and *CmoMLO18* are closely related and both belong to clade V, we examined their expression using real-time qPCR in the two squash genotypes. The relative expression patterns for both genes showed no significant differences between the genotypes at various time points (Figure 5B) and reflected the same trends as the FPKM values from the transcriptome data.

## 3. Discussion

### 3.1. Origin of Powdery Mildew Resistance of Squash

Powdery mildew significantly impacts the fruit quality and productivity of *C. moschata* and other commercial *Cucurbita* species [30,31]. The inherent genetic resistance of *Cucurbita* species to powdery mildew is occasionally detected in *C. moschata* and is usually absent in *C. pepo* [23,27,32]. Nevertheless, in both species, resistance has been effectively achieved by introgression of a key resistance locus, named *Pm-0*, obtained from a resistant wild *Cucurbita* species, *C. okeechobeensis* subsp. *Martinezii*, which is used widely in breeding and found in nearly all commercial powdery mildew resistance *C. moschata* and *C. pepo*. The locus *Pm-0* is considered responsible for resistance in nearly all powdery mildew resistance commercial cultivars of squash, pumpkin, and zucchini. Linkage mapping identified that the *Pm-0* locus within a 76.4 kb interval contained 14 candidate genes on LG10 [23]. Since the publication of the *C. moschata* genome [33], we found the locus *Pm-0* located on chromosome 3, CmoCh03: 7,539,988~7,620,164, based on the sequences of two flanking markers. Seventeen genes are located on the interval. According to our transcriptome data, only one gene, CmoCH03G010130, appeared in the list of 716 differentially expressed genes under non-inoculated conditions (Appendix A). CmoCH03G010130 is annotated as Cyclin-H1-1 with cyclin-dependent protein serine/threonine kinase regulator activity. This gene might have broad gene function rather than a direct connection with powdery mildew, as no publication has associated it with powdery mildew.

Other candidate genes in this interval that are highly likely to be involved in powdery mildew disease resistance include CmoCh03G010060, which is the homolog to At5G66900, an NBS-LRR protein in *Arabidopsis thaliana* that contains a domain similar to the *RPW8* locus known for conferring resistance to powdery mildew, and CmoCh03G010010, annotated as peroxidase 10. Additionally, CmoCh03G010100, annotated as a *salicylic acid-binding protein 2-like* (*SABP2*), has been identified in *Arabidopsis* for a role in powdery mildew resistance [34]. In our study, CmoCh03G010060 and CmoCh03G010010 did not show significant expression differences in the two squash genotypes from the start to 10 days post-inoculation. However, CmoCh03G010100, the *SABP2-like* gene, exhibited notable expression changes. Salicylic acid is crucial for the activation of defense genes in *Arabidopsis* [35,36], and high levels of SA are essential for activating local and systemic defense responses in cucumber and zucchini resistance to powdery mildew [37,38]. The expression of the *SABP2-like* gene is linked to the salicylic acid level, although it is unclear whether the *Pm-0* resistance locus has been introgressed into the two genotypes in our research from previous breeding efforts. The significant downregulation of *SABP2*-like gene expression at 6 h post-inoculation in the genotype SF02 might be related to powdery mildew susceptibility. An 11 bp insertion at the splice site region and two nonsynonymous mutations in the coding region of SF02 (as shown in Appendix A) may result in a loss of function of this gene, thereby affecting SA expression. In contrast, the YD26 genotype maintains high levels of SA expression, enhancing resistance to powdery mildew through various defense mechanisms. This suggests that the high expression level of *the SABP2-like* gene might act as the causal gene in the *Pm-0* locus, assuming the *Pm-0* resistance locus was introgressed into at least one of the two genotypes. Alternatively, the observed improvement in powdery mildew resistance may be attributed to broad-spectrum resistance mediated by salicylic acid.

### 3.2. Co-Expressed R Gene Cluster for Powdery Mildew Resistance

Powdery mildew resistance is considered a trait with a quantitative genetic basis in cucurbit plants [30]. In butternut squash, besides DEGs within the known resistance locus *Pm-0* on chromosome 3, we identified a group of disease resistance gene analogs (RGAs) on chromosome 6 that showed significant expression differences between the two genotypes. This was determined through transcriptome data and qPCR expression analysis. These R gene analogs, which include genes with CC, NBS, and LRR domains, belong to the CNL class [39]. The disease resistance proteins recognize specific pathogen effectors and then activate a series of defense responses, including triggering signal transductions, activating hypersensitive responses (HRs), producing and releasing immune-related hormones [36]. In melon, candidate gene resistance to *P. xanthii* was found located in an RGA-rich region, and this region also conferred resistance to aphids and viruses [40]. Two nearby genes conferring resistance to three races of powdery mildew pathogen were identifies as NBS-LRR genes [41]. Although R gene-mediated disease resistance often exhibits race specificity, meaning that when pathogens mutate or evolve into new races, the R genes may no longer provide effective resistance, it is still evident that the immune response triggered by R genes represents a crucial defense mechanism for plants against pathogen infection.

Plant resistance genes have to evolve in response to selective pressure for resistance to pathogens [42]. NBS-LRR resistance genes have a clustered nature due to gene duplication and when the resistance genes are closely related and physically linked, which would promote more R gene conversion or recombination [43,44,45]. The well-known *Mla* locus of powdery mildew resistance in barley is associated with several NBS-LRR gene homologs [4,5]. The allelic variations in *Mla* are due to InDel and SNP mutations. In wheat, powdery mildew resistance alleles of *Pm3* were found to evolve from the susceptible allele, and the mutations were all located in the LRR region [46]. In *Cucumis melo*, the *Pm-w* resistance locus, which confers high-level resistance to powdery mildew, was also found within a clustered NLRs region [47,48]. In our research, the high expression of RGAs transcripts in resistance genotype YD26 suggests the potential involvement of these LRR proteins in disease resistance to powdery mildew. This contrasts with the susceptible genotype SF02, where four frameshifts and two codon changes plus codon deletion within the R-gene cluster region (Appendix A) might lead to the lack of LRR protein expression. To further understand the role of these R genes, it will be necessary to identify the candidate locus from this region in an F2 population derived from the cross between YD26 and SF02.

### 3.3. The R and S Genes for Powdery Mildew Resistance

As an obligate biotrophic pathogen, powdery mildew relies on the living cells and tissues of host plants to obtain nutrients for growth and reproduction. After the spores of powdery mildew germinate, hyphae are produced, which invade the epidermal cells, forming haustoria and secondary hyphae to absorb nutrients from the host cells, and then new spores are produced to complete their life cycle [1]. To prevent the attack from powdery mildew, plant R gene-mediated resistances occur when plant R proteins recognize pathogen effectors inside the host cell after a successful invasion. The mechanisms of S gene-mediated resistances underlying host–pathogen interactions in plants are different. The first known S gene is barley *mlo*, which was found early in the 1930s [13]. The MLO proteins have a specific seven-transmembrane (TM) domain [18], and the recessive mlo allele protein prevents spore penetration by forming cell wall appositions at the penetration site [16].

Research shows that MLO proteins exist in all plants, and the origin of *MLO* genes can be traced back at least to the early evolutionary stages of land plant development approximately 400–450 million years ago [49]. The requirement for MLO proteins of powdery mildews to enter plant cells has been conserved for at least 200 million years, at or before the monocot–dicot split [50], and diverse powdery mildews take advantage of MLO protein on membrane over time. Since the discovery of *mlo*-resistance in barley, powdery mildew resistance arising from *MLO* allele mutations has also been found in monocot wheat, as well as in dicot plants such as *Arabidopsis*, tomato, melon, cucumber, etc. [16]. In *Arabidopsis*, *MLO* negatively regulates PEN1-dependent secretion and results in an early failure of the penetration process of pathogens [51].

The S and R genes represent the different mechanisms underlying plant resistance to powdery mildew. R gene-related resistance is usually considered to be readily overcome by the evolution of pathogens, while the *MLO* resistance used in spring barley widely for decades provides the best evidence that S gene-related resistance is durable. These S genes that would provide durable resistance have their own primary genetic functions, so the loss of function may result in undesirable side effects, such as increased sensitivity to other stresses, dwarfing, or other negative consequences [12]. Barley *MLO* mutants show increased susceptibility to the rice blast fungus, accelerated leaf senescence, and reduced grain yield [14]. Although the *MLO* gene has been discovered in other crops, it has not been widely applied in commercial breeding as it has been in barley. In this study, despite the difference in powdery mildew resistance between the two squash varieties, the expression of the two *MLO* genes belonging to clade V in both varieties was not suppressed. Consequently, neither of them formed an effective physical barrier, allowing the powdery mildew pathogen to invade the pumpkin leaves further and trigger the subsequent expression of downstream R genes response.

## 4. Materials and Methods

### 4.1. Plant Materials and Powdery Mildew Inoculation

Two squash genotypes, ‘YD26’ and ‘SF02’, were used in this research for powdery mildew inoculation. These genotypes were provided by the Vegetable Research Institute, Hainan Academy of Agricultural Sciences, and were identified through years of field observations in Hainan as resistant and susceptible to natural powdery mildew disease, respectively. Seeds of these genotypes were soaked in warm water at 50 °C to induce germination and then placed in a germination chamber. Once germinated and with developing roots, the seeds were transferred to Jiffy peat pellets and subsequently to 9 cm pots. The plantlets were grown in a chamber with a day/night temperature of 25/20 °C and a 16 h photoperiod.

Conidia of *P. xanthii* were collected directly from infected squash leaves and first inoculated onto the susceptible genotype ‘SF02’ to purify and propagate the pathogen. When seedlings of ‘YD26’ and ‘SF02’ had developed two true leaves, powdery mildew spores were collected and suspended in a freshly prepared solution of 0.5 × 10^6^ spores/mL with Tween-20 for spray inoculation. Squash leaves surface-inoculated with powdery mildew were marked for sampling at 06, 12, 24, 36, and 48 h and 10 days (240 h) post-inoculation (hpi). Non-inoculated leaves (00 hpi) of both genotypes were used for whole-genome sequencing. Additionally, three leaves from both non-inoculated and inoculated treatments of the two squash genotypes were sampled for transcriptome sequencing. On day 10, infected leaves were evaluated for pathogen infection and scored on a scale from 0 (no infection) to 5 (completely infected).

### 4.2. Whole Genome Re-Sequencing and Variation Identification of Squash

Genomic DNA was extracted from young leaves of two-week-old seedlings of the ‘YD26’ and ‘SF02’ genotypes using the CTAB method. The concentration and quality of the total genomic DNA were assessed with a NanoDrop 2000 Spectrophotometer (Thermo Fisher Scientific, Waltham, MA, USA) and gel electrophoresis. DNA libraries with 350 bp fragments were prepared for Illumina/BGI sequencing, and sequencing was performed on an Illumina platform by Biomarker Technologies in Beijing, China, with 150 bp read lengths. Raw reads were processed using the BMKCloud online platform. During quality control, clean data were generated by removing reads with adapters, poly-N sequences, and low-quality reads. Metrics such as Q20, Q30, GC content, and sequence duplication levels were calculated from the clean data. The clean sequencing reads were then mapped to the reference *C. moschata* genome for variation analysis.

### 4.3. Transcriptome Analysis after Powdery Mildew Inoculation

Two leaves of the squash genotypes ‘YD26’ and ‘SF02’ from each treatment were inserted in one tube and immediately submerged in liquid nitrogen. All samples were stored at −80 °C until used for transcriptome sequencing. The total RNA was extracted separately using the TIANGEN RNA kit following the manufacturer’s protocol. After checking and evaluating the quality, quantity, and integrity, RNA samples were used for transcriptome sequencing library preparation. The transcriptome sequencing library was constructed through RNA randomly fragmentation, cDNA strand 1/strand 2 synthesis, end repair, A-tailing, ligation of sequencing adapters, size selection, and library PCR enrichment, after which the library preparations were sequenced on an Illumina HiSeq 2000 platform and 150 bp paired-end reads were generated.

Raw reads in fastq format were processed and filtered to obtain the raw reads by moving the adapter and trimming the low-quality base. At the same time, Q20, Q30, and GC contents of the clean data were calculated. All the downstream analyses were based on the clean data with high quality. The *C. moschata* genome was used again, and the generated clean reads were mapped to the reference genome. Gene expression levels were quantified using Htseq-count, and FPKM values were calculated based on gene length and read counts, accounting for sequencing depth and gene length simultaneously. Gene Ontology (GO) and KEGG pathway enrichment analyses were conducted using the topGO and clusterProfiler R packages, respectively, to identify significantly enriched terms and pathways among differentially expressed genes.

### 4.4. Differentially Expressed Genes (DEGs) and Disease-Related Gene Identification

To identify genes potentially induced by powdery mildew infection, we compared transcriptome data across various groups based on their expression patterns. The analysis focused on contrasting the non-treatment groups of the two squash genotypes, ‘YD26’ and ‘SF02’, with different time points post-infection against their respective non-treatment controls. For differential expression analysis, DESeq2 was employed for datasets with biological replicates, while edgeR was used for datasets lacking biological replicates. Statistical significance was determined with a threshold of 0.05, and *p*-values were adjusted for multiple comparisons using the Benjamini and Hochberg method to control the false discovery rate. MLO protein sequences from *Arabidopsis*, cucumber, and monostylous crops (barley, rice, etc.) were obtained from NCBI aa. Blasting these *MLO* sequences from the public database with our butter squash sequencing data, twenty *C. moschata MLO* (short as *CmMLO*) sequences were identified and named *CmMLO1* to *CmMLO20*, respectively, based on the location on the chromosome of *C. moschata*. A phylogenetic tree was constructed by MEGA 7.0.26 with the maximum likelihood method and 1000 bootstrap replications using all the MLO protein sequences from *Arabidopsis*, cucumber, and monocotylous crops.

### 4.5. Identification of Gene Expression Analysis

Two leaves from infected squash seedlings were sampled for total RNA extraction. Three biological replicates were used for each treatment, and the same samples were employed for RNA-seq. RNA was extracted using Trizol (Invitrogen, Carlsbad, CA, USA) following the manufacturer’s protocol. The quality and quantity of RNA samples were assessed using a 1% agarose gel and a NanoDrop spectrophotometer. Following DNase treatment to remove genomic DNA, 1 µg of RNA was used for cDNA synthesis using the iScript™ cDNA Synthesis Kit (Bio-Rad, Hercules, CA, USA). For quantitative real-time PCR (qRT-PCR), a 10 µL reaction mix was prepared, consisting of 2 µL cDNA template (~50 ng/µL), 5 µL 2× iQ™ SYBR^®^ Green Supermix (Bio-Rad, CA, US), 0.3 µL each of forward and reverse primers (10 µM), and 2.7 µL Milli-Q water. The qRT-PCR was conducted with an initial denaturation step at 95 °C for 3 min, followed by 40 cycles of 15 s at 95 °C and 1 min at 60 °C. A final melting curve was generated by holding at 95 °C for 10 s, then at 65 °C for 5 s, with a gradual increase to 95 °C at 0.5 °C increments. ACTIN was used as the reference gene for normalization, and the primer sequences are listed in Appendix A.

## Figures and Tables

**Figure 1 ijms-25-10896-f001:**
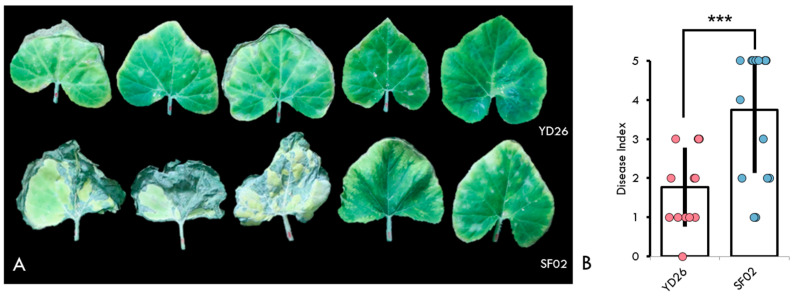
Symptoms and disease scoring of powdery mildew infection. (**A**) Visible symptoms of powdery mildew infection on leaves on day 10 of evaluation. The top row shows five leaves from genotype YD26, and the bottom row shows five leaves from genotype SF02. (**B**) Bar chart representing the average disease index for genotypes YD26 and SF02. Dots on the chart indicate individual plant scores within each genotype. *** indicate statistical significance at *p* < 0.001.

**Figure 2 ijms-25-10896-f002:**
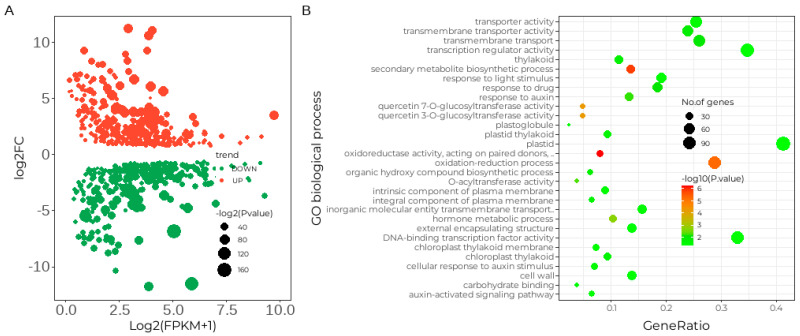
Differentially expressed genes before powdery mildew infection (at 0 hpi). (**A**) Scatter plot showing the 716 DEGs with upregulated trends (red dots) and downregulated trends (green dots) in YD26 versus SF02. (**B**) Annotation of 716 differentially expressed genes.

**Figure 3 ijms-25-10896-f003:**
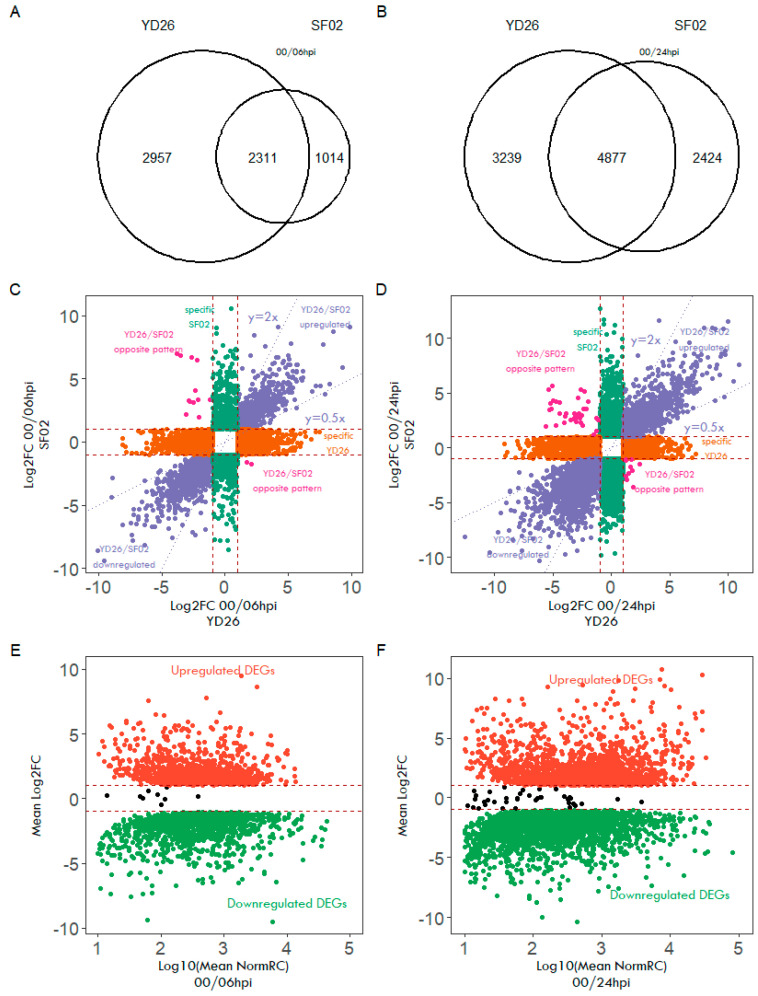
Differentially expressed genes in response to powdery mildew infection. (**A**) Venn diagram showing the overlap of differentially expressed genes in genotypes YD26 and SF02 at 06 hpi compared to non-infection (00 hpi). (**B**) Venn diagram showing the overlap of differentially expressed genes in YD26 and SF02 at 24 hpi compared to non-infection. (**C**) Scatter plot comparing Log2 fold change (Log2FC) values of differentially expressed genes at 06 hpi between genotypes YD26 and SF02. The y = 2x line indicates that the Log2FC values for DEGs in SF02 are twice as high as those in YD26, while the y = 0.5x line indicates that the Log2FC values for DEGs in YD26 are twice as high as those in SF02. Pink dots represent DEGs with opposite expression patterns between the two genotypes. (**D**) Scatter plot comparing Log2FC values of DEGs at 24 hpi in YD26 and SF02. (**E**) Scatter plot representing mean Log2FC and mean normalized Read counts (RCs) for the shared 2311 DEGs with upregulated (red dots) or downregulated (green dots) patterns in response to powdery mildew infection at 06 hpi compared to non-infection (00 hpi). Black dots indicate the DEGs with opposite expression patterns in the two genotypes at 06 hpi. (**F**) Scatter plot representing mean Log2FC and mean normalized RC for the shared 4877 DEGs with upregulated (red dots) or downregulated (green dots) patterns in response to powdery mildew infection at 24 hpi compared to non-infection (00 hpi). Black dots indicate the DEGs with opposite expression patterns in the two genotypes at 24 hpi.

**Figure 4 ijms-25-10896-f004:**
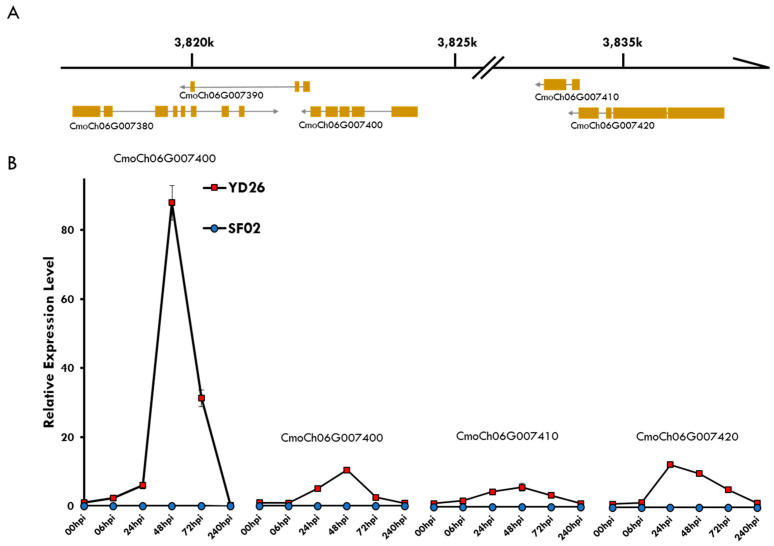
The location and expression of R genes on chromosome 6 of *C. moschata*. (**A**) Location and gene structure of five nearby genes on chromosome 6 of *C. moschata*, spanning the region CmoCh06: 3,817,738~3,836,887. (**B**) Normalized relative expression levels of these nearby RGA genes, as determined by qRT-PCR.

**Figure 5 ijms-25-10896-f005:**
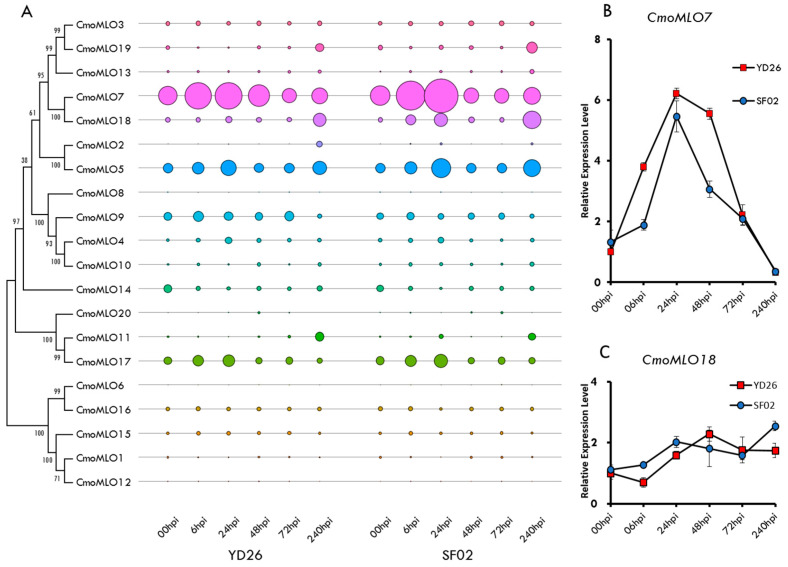
(**A**) The phylogeny tree of MLO-like proteins and the FKPM values of MLO-like genes in *C. moschata*. Bubble sizes correspond to transcript expression levels in response to powdery mildew infection at different time points, with larger bubbles indicating higher expression levels. (**B**) The qRT-PCR expression levels of *CmoMLO7* from clade V in *C. moschata* genotypes YD26 and SF02. (**C**) The qRT-PCR expression levels of *CmoMLO18* from clade V in *C. moschata* genotypes YD26 and SF02.

## Data Availability

The original contributions presented in the study are included in the Appendix A, further inquiries can be directed to the corresponding author/s.

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
