# Peer review of "Identification of Powdery Mildew Resistance-Related Genes in Butternut Squash (Cucurbita moschata)"

_ijms, 2024, doi:10.3390/ijms252010896_

Round 1

Reviewer 1 Report

Comments and Suggestions for Authors

Powdery mildew is one of the most important pathogens that significantly affects production and quality of butternut squash. In this paper titled “Identification of powdery mildew resistance related genes in butternut squash (Cucurbita moschata) (ijms-3182476), transcriptome and genome sequencing analysis were used to identify SNPs and potential genes associated with powdery mildew resistance in butternut squash, and analysis of differentially expressed genes (DEGs) following powdery mildew infection has revealed several genes involved in resistance. These findings are of

 important scientific value.

Comments:  

(1) For 2.2. Variations from whole genome and transcriptome sequencing, the SNP variations between the two genotypes YD26 and SF02 were analyzed roughly, which was not closely related to topic of the manuscript. It would be better to further analyze the SNPs present in DEGs, which may be related to powdery mildew resistance.

(2) The Figures and Tables should be numbered in the order they appear in the text.

(3) Line 124, Supplementary Table S2 should be Supplementary Table S1? Please check it.

(4) Line 230, Supplementary Table S3 should be Supplementary Table S3? Please check it.

Author Response

Dear Reviewer,

Thanks for your time, and we modified our manuscript based on all your comments.

(1) For 2.2. Variations from whole genome and transcriptome sequencing, the SNP variations between the two genotypes YD26 and SF02 were analyzed roughly, which was not closely related to topic of the manuscript. It would be better to further analyze the SNPs present in DEGs, which may be related to powdery mildew resistance.

Thank you for pointing out! We added a sentence in 2.2 "In addition to the overall differences between the two genotypes, the following study conducted a detailed analysis of genetic variations within specific potential candidate genes that might contribute to the observed differences."

And also in the discussion part, we discussed the specific variations in the SABP2-like gene and R gene cluster with additional Supplemental Figures S5 and S7.

(2) The Figures and Tables should be numbered in the order they appear in the text.

(3) Line 124, Supplementary Table S2 should be Supplementary Table S1? Please check it.

(4) Line 230, Supplementary Table S3 should be Supplementary Table S3? Please check it.

Thanks for your comments. We double-checked and renumbered all the Figures and Tables numbers in order.

Reviewer 2 Report

Comments and Suggestions for Authors

In this study, Fu et al. analyzed SNPs and potential powdery mildew resistance genes in butternut squash using transcriptome and genome sequencing data. My comments are as follows:

In the Introduction, the authors cited the results of mlo-based powdery mildew resistance in barley, however, there may be big differences between barley and Cucurbit crops, the authors did not mention the similarity of this protein in these two different plants. it would be better to introduce the results of mlo “in dicot plants such as Arabidopsis, tomato, melon, and cucumber etc” in Introduction.

“Several genetic studies have identified loci associated with resistance to powdery mildew in Cucurbita”, for these important results, it is too simple to describe like this using only one sentence.

“Statistical analysis confirmed the significant differences in powdery mildew resistance between YD26 and SF02 on day 10 (240hpi)”, “both genes showed no significant differences between the genotypes” results needs to be cited.

The authors used a threshold of p-values < 0.05 to select the DEGs and found thousands of DEGs, however, without considering the foldchanges is meaningless.

The Supplementary Table numbers are not in order.

The authors did not show the verification of the transcriptome data with qRT-PCR.

Error bars need to be added in all the figures showing a transcriptional level.

minor comments:

there are many Grammarly mistakes, e.g. Line 30 “This region is a major producer”, “T-test”, and also maybe there is something wrong with their address.

Comments on the Quality of English Language

Grammar mistakes.

Author Response

Dear reviewer,

Thank you very much to review our manuscript, and we modified the paper by your comments as follow:

comments 1. In the Introduction, the authors cited the results of mlo-based powdery mildew resistance in barley, however, there may be big differences between barley and Cucurbit crops, the authors did not mention the similarity of this protein in these two different plants. it would be better to introduce the results of mlo “in dicot plants such as Arabidopsis, tomato, melon, and cucumber etc” in Introduction.

we added a paragraph after the introduction of mlo-based PM resistance in barley.

As more genomes were sequenced, it became clear that MLO genes are part of a large gene family, all encoding seven-transmembrane proteins, and have diversified into several clades with different functions in plants[17, 18]. The mlo genes involved in powdery mildew resistance in monocots like barley and wheat are in clade IV, while in dicots, they belong to clade V[17]. The MLO gene also found played an important role in response to PM in cucumbers and other cucurbit crops. In cucumber, a MLO (Csa5G623470) in clade V, linked to the locus pm5.1, has been identified as the causal gene for durable powdery mildew resistance[19]. The same loss-of-function mutation MLO genes (Csa5G623470, named CsaMLO8) was proved the cause of hypocotyl resistance to PM[20]. Other two MLO genes in clade V were also cloned, and overexpression of all three CsaMLO genes in the tomato mlo mutant restored susceptibility to PM with a different extent[21].

comments 2. “Several genetic studies have identified loci associated with resistance to powdery mildew in Cucurbita”, for these important results, it is too simple to describe like this using only one sentence.

we added after this sentence:

Among these, the Pm-0 locus which was considered that introgressed from a wild Cucurbita species, provides resistance to powdery mildew in commercially cultivated Cucurbita varieties[23, 28]. This locus contains several candidate genes might together contribute the resistance[23]. Another major dominant locus CpPM10.1 in zucchini contains an Arabidopsis RPW8 domain after fine mapping[27]. The MLO genes in clade V of Cucurbita species were identified and shown to exhibit differential expression between resistance and susceptible plants[29, 30]. 

comments 3. “Statistical analysis confirmed the significant differences in powdery mildew resistance between YD26 and SF02 on day 10 (240hpi)”, “both genes showed no significant differences between the genotypes” results needs to be cited.

we added :(Figure 1B);(Figure 5B)

comments 4. The authors used a threshold of p-values < 0.05 to select the DEGs and found thousands of DEGs, however, without considering the foldchanges is meaningless.

we added "and an absolute log2FC > 0.5"

comments 5. The Supplementary Table numbers are not in order.

we renumbered the supplementary files.

comments 6. The authors did not show the verification of the transcriptome data with qRT-PCR.

we just focused on the two MLO genes in clade V and the four R gene in cluster with qRT-PCR. 

comments 7. Error bars need to be added in all the figures showing a transcriptional level.

we added the error bars on Figure 4B and 5B for the qPCR results

comments 8. minor comments:

there are many Grammarly mistakes, e.g. Line 30 “This region is a major producer…”, “T-test”, and also maybe there is something wrong with their address.

we modified the sentences.

It is the main vegetable production area including butternut squash for the mainland Chinese market, particularly during winter.

A significant difference in disease index was observed between YD26 and SF02, with YD26 exhibiting notably higher resistance, as confirmed by a t-test analysis (p = 0.00077)

address changed to:

1   Beijing Vegetable Research Center (BVRC), Beijing Academy of Agriculture and Forestry Science, Beijing 100097, China

2   National Engineering Research Center for Vegetables (NERCV), State Key Laboratory of Vegetable Biobreeding, Beijing Academy of Agriculture and Forestry Sciences, Beijing 100097, China

3   Vegetable Research Institute, Hainan Academy of Agricultural Sciences, Haihou 571199, China

Reviewer 3 Report

Comments and Suggestions for Authors

Please see the returned marked-up document for editing suggestions.   Overall, nice work.  

Comments on the Quality of English Language

see attached

Author Response

Dear Reviewer,

Thanks for your time on our manuscript, we modified all based on your comments, please check the new version.
